# Effective Cryopreservation of a Bioluminescent Auxotrophic *Escherichia coli*-Based Amino Acid Array to Enable Long-Term Ready-to-Use Applications

**DOI:** 10.3390/bios11080252

**Published:** 2021-07-26

**Authors:** Hee Tae Ahn, In Seung Jang, Thinh Viet Dang, Yi Hyang Kim, Dong Hoon Lee, Hyeun Seok Choi, Byung Jo Yu, Moon Il Kim

**Affiliations:** 1Department of BioNano Technology, Gachon University, 1342 Seongnamdae-ro, Sujeong-gu, Seongnam 13120, Korea; venice4@naver.com (H.T.A.); dvietthinh96@gmail.com (T.V.D.); dhlee9219@gmail.com (D.H.L.); 2Green and Sustainable Materials R&D Department, Research Institute of Clean Manufacturing System, Korea Institute of Industrial Technology (KITECH), Cheonan 31056, Korea; isjang@kitech.re.kr (I.S.J.); kyscent@kitech.re.kr (Y.H.K.); hchoi@kitech.re.kr (H.S.C.)

**Keywords:** long-term preservation, cryoprotectant, *Escherichia coli* auxotroph, amino acid array, trehalose

## Abstract

Amino acid arrays comprising bioluminescent amino acid auxotrophic *Escherichia coli* are effective systems to quantitatively determine multiple amino acids. However, there is a need to develop a method for convenient long-term preservation of the array to enable its practical applications. Here, we reported a potential strategy to efficiently maintain cell viability within the portable array. The method involves immobilization of cells within agarose gel supplemented with an appropriate cryoprotectant in individual wells of a 96-well plate, followed by storage under freezing conditions. Six cryoprotectants, namely dimethyl sulfoxide, glycerol, ethylene glycol, polyethylene glycol, sucrose, and trehalose, were tested in the methionine (Met) auxotroph-based array. Carbohydrate-type cryoprotectants (glycerol, sucrose, and trehalose) efficiently preserved the linearity of determination of Met concentration. In particular, the array with 5% trehalose exhibited the best performance. The Met array with 5% trehalose could determine Met concentration with high linearity (R^2^ value = approximately 0.99) even after storage at −20 °C for up to 3 months. The clinical utilities of the Met and Leu array, preserved at −20 °C for 3 months, were also verified by successfully quantifying Met and Leu in spiked blood serum samples for the diagnosis of the corresponding metabolic diseases. This long-term preservation protocol enables the development of a ready-to-use bioluminescent *E. coli*-based amino acid array to quantify multiple amino acids and can replace the currently used laborious analytical methods.

## 1. Introduction

The basic building blocks of proteins are amino acids, which have important roles in various physiological processes. The dysregulation of amino acid metabolism can lead to pathological conditions, including nutritional imbalances and metabolic disorders [1,2]. Therefore, the development of a reliable, accurate, and sensitive analytical method to determine amino acids in biofluids such as blood or urine has piqued the interest of the scientific community. Conventional analytical techniques, such as high-performance liquid chromatography, mass spectrometry, fluorometry, and immunoassays are currently being used [3,4,5,6]. The high sensitivity of conventional analytical techniques enables the precise quantification of amino acids. However, these analytical techniques are associated with several limitations, such as complex pre-treatment and post-treatment steps, inefficient labeling, instability of reagents, and the need for skilled technicians to operate expensive and complex instrumentation, which are associated with high analysis costs and time-consuming processes [6]. Therefore, there is a need to develop an effective, easy, rapid, economical, and reliable analytical method for amino acids. 

Microbial cell-based assays are potential alternatives to the currently employed analytical methods [7,8,9,10]. These assays are based on the unique growth behavior of reporter cells, which enables the convenient detection of target amino acids at a low cost. Unmodified bacteria can be utilized as reporter cells in these assays. However, genetically engineered bacterial cells have been examined to generate dose-dependent signals of diverse amino acids [11]. Among the various microorganisms used in these assays, auxotrophic mutants of *Escherichia coli* are reported to be efficient systems for the quantification of amino acids [12]. Additionally, bioluminescence-based measurement of bacterial responses is an efficient technology that allows rapid, quantitative, and non-invasive detection of microbial populations [13,14]. Bioluminescence has been conventionally generated from luciferase-catalyzed oxidation of luciferin; however, this strategy should involve additional exogenous luciferin that is hard to penetrate through the cell membrane. On the other hand, bacterial bioluminescence is efficient since the cells can produce bioluminescence by utilizing flavin mononucleotides present abundantly within the cells [15]. Amino acids in the samples dose-dependently promote the growth of auxotrophic *E. coli*. Thus, the concentration of amino acids in the sample is determined based on the cell growth-dependent luminescence response. According to the potential of cell-based amino acid assays, we previously developed an amino acid diagnostic array, which comprised 16 different *E. coli* amino acid auxotrophs within the agarose gel in individual wells of a 96-well plate. The cells in the array exhibited rapid, specific, and sensitive growth depending on the concentration of the test amino acids [16]. The amino acid array was used for the sensitive and specific detection of 16 different amino acids by measuring the luminescence intensity within 4 h. Additionally, the amino acid array was applied to determine the concentrations of other important biomarkers, including homocysteine and galactose [17,18]. 

The *E. coli* auxotroph-based amino acid array is comprised of the following two main components: a 96-well plate and cells [19]. The cells were maintained in a frozen state and delivered through cold-chain transportation. To prepare the amino acid array, the end users must integrate the two components through several preparatory steps, including thawing, seeding, culturing, centrifuging, and counting the cells, followed by the preparation of the cell array [20]. The amino acid array preparation involved several days of cell culture. Additionally, the operations of this array are limited only to facilities with established cell culture infrastructure and experienced technicians [21]. Furthermore, cell array preparation may be associated with potential variations, which may lead to run-to-run variability and poor performance. To overcome the limitations of delivering each component to the end users separately, the entire amino acid array system can be prepared at the manufacturing stage and transported to the end users in a ready-to-use format [22]. However, in this case, the cells within the array should be carefully preserved to maintain their viability during storage and transportation [23]. Therefore, the storability of the cells is critical for the practical applications of the amino acid array platform. 

Here, we developed a ready-to-use amino acid array comprising amino acid auxotrophic *E. coli* immobilized within agarose gel supplemented with an optimized cryoprotectant in individual wells of a 96-well plate. The developed amino acid array could be efficiently stored in a simple freezer (−20 °C) for up to 3 months and used for analysis by adding a sample solution to each well of the array. The amino acid array enabled determination of the target amino acid concentration with high linearity, yielding an R^2^ value of approximately 0.99, after storage for up to 3 months. This study screened for an optimal cryoprotectant and examined its effect on the quantitative determination of the target amino acid. 

## 2. Materials and Methods

### 2.1. Materials

Amino acid auxotrophs for methionine (Met), isoleucine (Ile), lysine (Lys), and leucine (Leu) harboring the pTAC-luc plasmid were prepared following a previously reported protocol [16]. Briefly, a tac promoter was added into the pETDuet-1 vector (Novagen, San Diego, CA, USA) to replace the T7lac promoter and generate the pTAC. The luc component was amplified using the pGL-Basic vector (Promega, WI, USA) and the PCR products were cloned into the pTAC to produce pTAC-luc. Finally, the pTAC-luc plasmids were transferred into competent cells by electroporation using a Gene Pulser system (BioRad, CA). M9 minimal medium, Luria-Bertani (LB) broth, ampicillin, kanamycin, isopropyl-β-D-thiogalactoside (IPTG), cyanocobalamin (vitamin B_12_), dimethyl sulfoxide (DMSO), glycerol, ethylene glycol (EG), polyethylene glycol (PEG), sucrose, trehalose, agarose (low gelling temperature), and amino acids (Met, Ile, Lys, and Leu) were purchased from Sigma-Aldrich (St. Louis, MO, USA). All other reagents and chemicals used in this study were of analytical grade.

### 2.2. Construction of Ready-to-Use Amino Acid Array Containing a Cryoprotectant

To prepare the Met array, *E. coli* Met auxotrophic strains were cultured in LB medium containing ampicillin (50 μg/mL) and kanamycin (50 μg/mL) at 37 °C overnight in a shaking incubator. The culture was centrifuged at 8000 rpm and 4 °C for 5 min. The cell pellet was washed twice with M9 minimal medium and seeded (2 × 10^6^ cells/well) into individual wells of a 96-well plate (Nunc, Roskilde, Denmark). Next, the cells (50 μL) were mixed with 3% agarose (50 μL) containing 5% or 10% (*w/v*) cryoprotectant (DMSO, glycerol, EG, PEG, sucrose, or trehalose) at a ratio of 1:1 (*v/v*). The mixture was allowed to solidify at room temperature (RT) for 20 min. The prepared ready-to-use amino acid array was sealed with parafilm, covered with aluminum foil, and stored at −20 °C and −80 °C until use. The Ile, Lys, and Leu arrays were prepared following the same procedures using Ile, Lys, and Leu auxotrophic strains of *E. coli*, respectively.

### 2.3. Evaluation of Cryopreservation Efficiency and Quantitative Determination of Amino Acids Using the Ready-to-Use Amino Acid Array

At a predetermined time point after storage at −20 °C and −80 °C, the ready-to-use Met array containing a cryoprotectant was defrosted for 1 h. The amino acid cocktail solution dissolved in M9 medium (100 μL) supplemented with 1 nM cyanocobalamin, 1 mM IPTG, and various concentrations of Met (0, 1.5625, 3.125, 6.25, 12.5, and 25 μM) was added to each well. Subsequently, the cell array was incubated at 37 °C for 4 h. Luminescence was measured using a microplate reader (Synergy H1, BioTek, VT). Scanning images were captured using a chemiluminescence imaging system (Alliance Q9 Mini, UVITEC, Cambridge, UK). The R^2^ value was calculated from the linear calibration curves of Met (plotted as concentrations vs. luminescence intensities). 

To evaluate the detection precision of the ready-to-use amino acid array after long-term storage, the amino acid arrays containing Met, Ile, Lys, and Leu auxotrophic strains of *E. coli* with 5% trehalose as a cryoprotectant were stored at −20 °C for 3 months. Thereafter, the amino acid arrays were defrosted, and amino acid cocktail solutions containing the corresponding target amino acid at 10 μM concentration were added to individual wells of the 96-well plate. The array was processed as described above. The concentrations of the target amino acids were determined from five independent experiments. The precision and reproducibility of the assays were assessed by determining the recovery rate (recovery [%] = measured value/actual value × 100) and the coefficient of variation (CV [%] = standard deviation/average × 100).

To evaluate the clinical utility of the ready-to-use amino acid array after long-term storage, the amino acid arrays containing Met and Leu auxotrophic strains of *E. coli* with 5% trehalose were stored at −20 °C for 3 months and applied to the clinical solutions containing the corresponding target amino acids (Met and Leu) at normal and patient level spiked in blood serum samples. The array was processed as described above. 

## 3. Results and Discussion

In this study, an amino acid array comprising amino acid auxotrophic *E. coli* harboring a bioluminescence plasmid was constructed for rapid and convenient quantification of multiple amino acids. As described in a previous protocol [16], amino acid auxotrophs for Met were constructed using the transposon mutagenesis method based on the selected *E. coli* (ATCC1105). The autotrophs were supplemented with the bioluminescence-generating plasmid pTAC-luc, constructed by inserting the luciferase gene (luc) derived from *Photinus pyralis* (firefly) into PET-pTAC containing an IPTG-inducible promotor and transformed to the auxotroph. As a result, the auxotrophs produced a bioluminescence signal without the involvement of any substrate during the growth in response to the presence of target amino acids. For practical ready-to-use applications, the cells on the array need to be stored for a sufficiently long period of time while maintaining their viability to hold the inherent analytical qualities. To this end, we tested several selected cryoprotectants that could provide extracellular and intracellular protection to the cells during storage under freezing conditions [24] to enhance the long-term storability and maintain the quantitative determination capability of the array for target amino acids. Although several studies have reported the benefits of cell cryopreservation [23,25], the cryopreservation of microbial cell-based amino acid arrays has not been previously reported. Six cryoprotectants (DMSO, glycerol, EG, PEG, sucrose, and trehalose), which have been widely utilized as cryoprotectants for preserving the cell viability, were added to individual wells of a 96-well plate containing Met auxotroph cultures in agarose; the mixture was then allowed to solidify at room temperature. The Met array containing the cryoprotectant was sealed and stored at −20 °C and −80 °C until use. For the analysis, a sample solution containing Met was added to individual wells after defrosting the array. The mixture was incubated for 4 h at 37 °C. Thereafter, the luminescence responses to various concentrations of Met in the samples were monitored. The amino acid array containing the optimal cryoprotectant was hypothesized to serve as an efficient cell-based amino acid biosensor with practical applications for determining the concentrations of target amino acids even after long-term storage (up to 3 months) (Figure 1 and Appendix A). 

The effects of six cryoprotectants added to the Met array on the linearity of determining Met concentration during the 16-day storage period were determined (Figure 2). The R^2^ values of the Met calibration curves generated using the control amino acid array (without cryoprotectant) decreased significantly (below 0.5) with the increase in storage time, indicating that the quantification ability of the amino acid arrays decreased in the absence of a cryoprotectant upon storage under freezing conditions. However, the R^2^ values of the Met calibration curves generated using the arrays containing cryoprotectants were markedly higher than those of the Met calibration curves generated using the control arrays. Different storage conditions, including cryoprotectant levels (5% and 10% (*w/v*)), freezing temperatures (−20 °C and −80 °C), and agarose immobilization (the presence or absence of agarose), were examined to determine the optimal storage and preparation conditions for the amino acid array. At a concentration of 5%, carbohydrate-based cryoprotectants (glycerol, sucrose, and trehalose) were efficient in preserving the linearity of Met detection.

Further screening was performed after storing the Met array at −20 °C or −80 °C for up to 3 months. The R^2^ values of the Met calibration curve generated using these Met arrays containing carbohydrate-type cryoprotectants were examined at predetermined time points (Figure 3). Compared with those of Met calibration curves generated using the control arrays stored at −20 °C, the R^2^ values of Met calibration curves generated using the control arrays (without cryoprotectant) stored at −80 °C were higher. However, the R^2^ values of Met calibration curves generated using the control amino acid arrays stored at −80 °C significantly decreased at day 90 (approximately 0.6) (Figure 3b). All three carbohydrate-type cryoprotectants preserved the linearity of Met detection. In particular, the arrays containing trehalose exhibited the best performance and yielded near-perfect linearity of Met detection under all test conditions. Although the R^2^ values of Met calibration curves were efficiently maintained after long-term storage for up to 3 months, the luminescence intensities of the samples containing 25 μM Met significantly decreased to less than half of the initial value at all storage conditions (Figure 4). This result may be attributed to the cryo-injury or osmotic damage of the cells [24]. The array containing trehalose exhibited the best performance and the highest luminescence intensities, although the luminescence intensity significantly decreased. Thus, 5% trehalose was selected as the optimal cryoprotectant for the amino acid array. The solid-phase array is easy to handle by end users. Additionally, storage in a simple freezer (−20 °C) is more convenient than that in a deep freezer (−80 °C). Hence, agarose-based immobilization and storage at −20 °C were selected as the preferable conditions for preparation of the array. 

The luminescence images of the Met arrays containing three carbohydrate-type cryoprotectants (glycerol, sucrose, and trehalose) and the Met calibration curves generated using these arrays before and after storage for 3 months are presented in Figure 5. The concentration of Met exhibited a linear positive correlation with luminescence intensity before and after storage of the array for 3 months, indicating that the amino acid array can accurately quantify Met even after storage for up to 3 months. Interestingly, in individual wells of the Met array stored for 3 months, several regions exhibited enhanced luminescence, which can be attributed to the aggregation of the viable cells that results in the concentration of luminescence to the corresponding regions within the well. 

The optimized cryoprotectant (trehalose at 5%) was added to the Ile, Lys, and Leu arrays to examine the general applicability of this strategy to efficiently preserve auxotrophic *E. coli*-based amino acid arrays. The results of the five independent quantitative experiments examining the concentrations of Met, Ile, Lys, and Leu (initial concentration: 10 μM) using the array stored for 3 months revealed accurate and precise determinations of amino acid concentrations with CV values and recovery rates in the ranges of 2.71–5.41% and 97.63–103.37%, respectively (Table 1), demonstrating the excellent reproducibility and reliability of the assay. These results provide evidence that the solid-phase amino acid array prepared with trehalose can be stored and used efficiently by end users for a sufficient time up to 3 months [26], demonstrating its benefits for realizing practical and commercial applications.

The most promising application of amino acid quantification in the clinical field is the diagnosis of aminoacidopathies including homocystinuria and maple syrup urine disease, which are associated with the high levels of Met and Leu, respectively, in human blood [27]. Thus, the diagnosing capability of the ready-to-use amino acid array, that was stored for 3 months, was demonstrated by determining normal (Met: 20 μM and Leu: 100 μM) and patient levels (Met: 100 μM and Leu: 300 μM) in artificial spiked blood serum samples [28]. Although a number of assays to determine amino acid concentrations in biological fluid have been developed, their long-term storability has rarely been studied [29]. As a result, a large difference between normal and patient samples was detected by the corresponding luminescence intensities, and furthermore, precise determination of different levels of Met and Leu were successfully achieved (Figure 6). These observations show that the ready-to-use amino acid array may serve as a promising analytical tool enabling diagnosis of aminoacidopathies in the clinic. 

## 4. Conclusions

This study demonstrated that trehalose is an effective cryoprotectant for the preservation of the auxotrophic *E. coli*-based amino acid array for up to 3 months at −20 °C without the loss of linearity of the target amino acid detection. The amino acid array containing trehalose as a cryoprotectant precisely, accurately, and conveniently quantified the concentrations of target amino acids even after storage for 3 months, presumably due to its effective prevention of cryo-injury and osmotic damage under the freezing storage condition. The cryo-array system developed in this study is a “ready-to-use” platform with potential practical applications in the development of microbial cell-based biosensors. 

## Figures and Tables

**Figure 1 biosensors-11-00252-f001:**
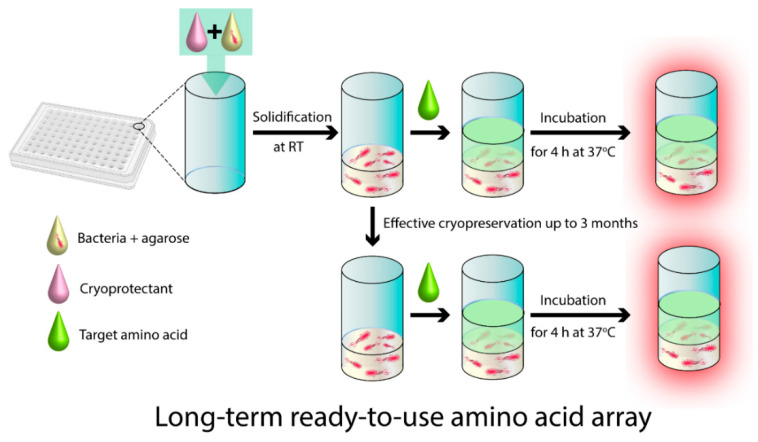
Schematic illustration of the ready-to-use amino acid array comprising amino acid auxotrophic bioluminescent *Escherichia coli* immobilized within the agarose gel and supplemented with an optimal cryoprotectant in individual wells of a 96-well plate. The amino acid array can be defrosted at any time during the cryopreservation period and used for the analysis of amino acids.

**Figure 2 biosensors-11-00252-f002:**
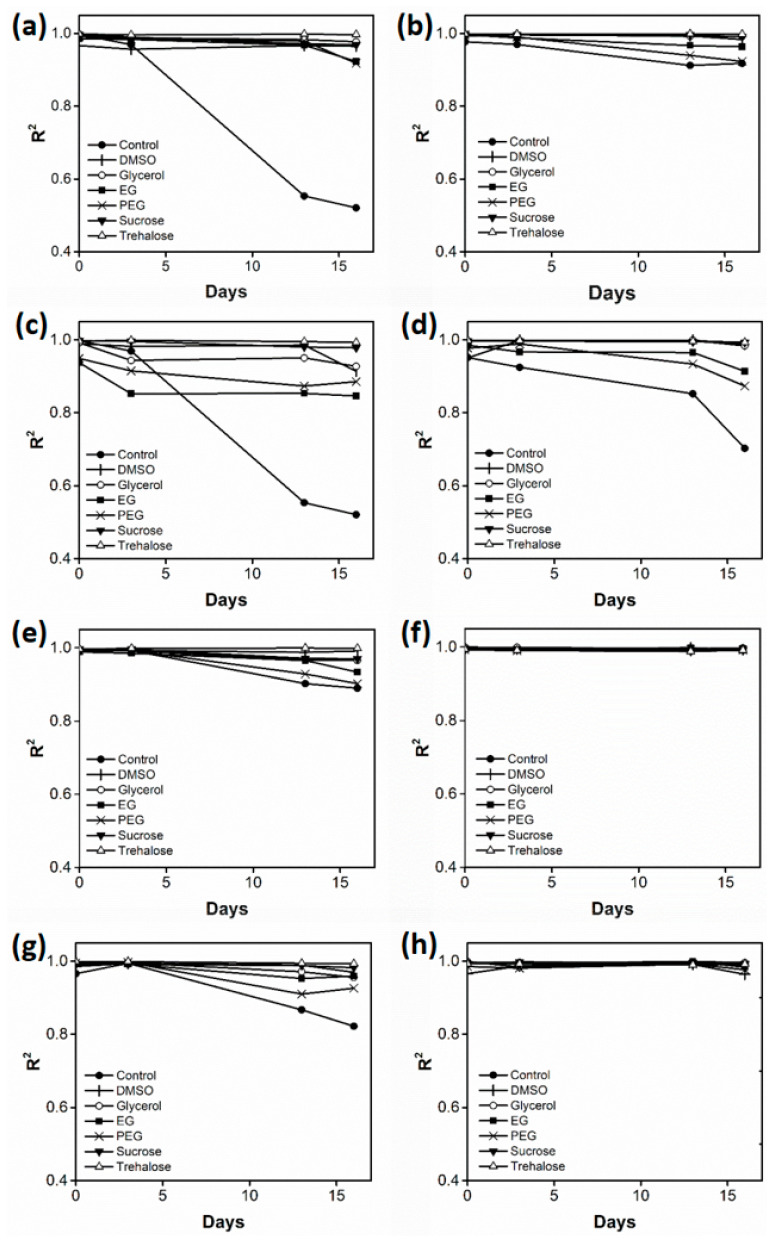
Coefficient of determination (R^2^) values of the Met calibration curves generated using the amino acid array during the 16-day storage period under the following conditions: (**a**) 5% cryoprotectant with agarose immobilization at −20 °C, (**b**) 5% cryoprotectant with agarose immobilization at −80 °C, (**c**) 10% cryoprotectant with agarose immobilization at −20 °C, (**d**) 10% cryoprotectant with agarose immobilization at −80 °C, (**e**) 5% cryoprotectant without agarose immobilization at −20 °C, (**f**) 5% cryoprotectant without agarose immobilization at −80 °C, (**g**) 10% cryoprotectant without agarose immobilization at −20 °C, and (**h**) 10% cryoprotectant without agarose immobilization at −80 °C.

**Figure 3 biosensors-11-00252-f003:**
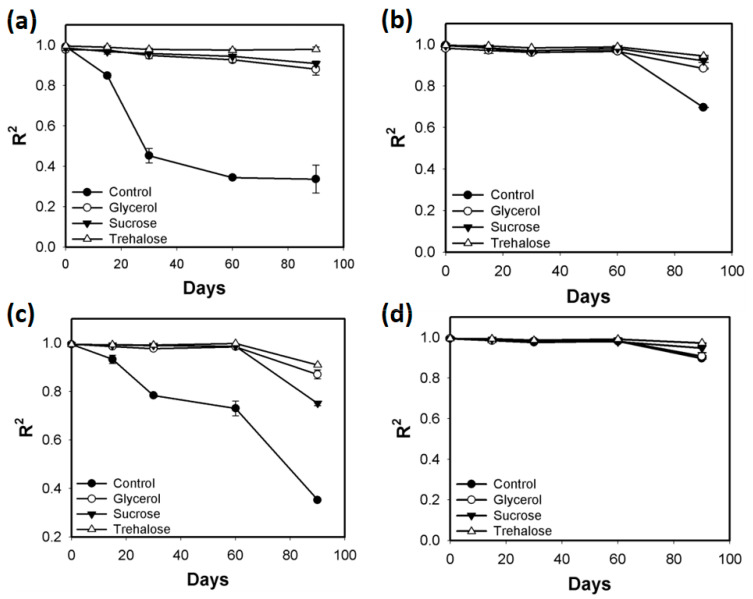
Coefficient of determination (R^2^) values of the Met calibration curves generated using the amino acid array containing carbohydrate-type cryoprotectants (5%) during the long-term storage period (3 months) under the following conditions: (**a**) agarose immobilization at −20 °C, (**b**) agarose immobilization at −80 °C, (**c**) without agarose immobilization at −20 °C, and (**d**) without agarose immobilization at −80 °C. Error bars represent standard errors derived from three independent measurements.

**Figure 4 biosensors-11-00252-f004:**
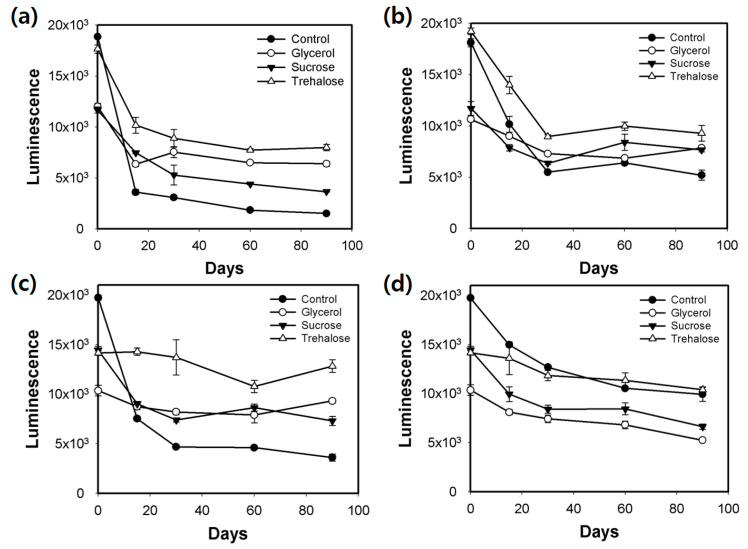
Luminescence intensities of the amino acid array containing carbohydrate-type cryoprotectants (5%) during the long-term storage period (3 months) under the following conditions: (**a**) agarose immobilization at −20 °C, (**b**) agarose immobilization at −80 °C, (**c**) without agarose immobilization at −20 °C, and (**d**) without agarose immobilization at −80 °C. Error bars represent standard errors derived from three independent measurements.

**Figure 5 biosensors-11-00252-f005:**
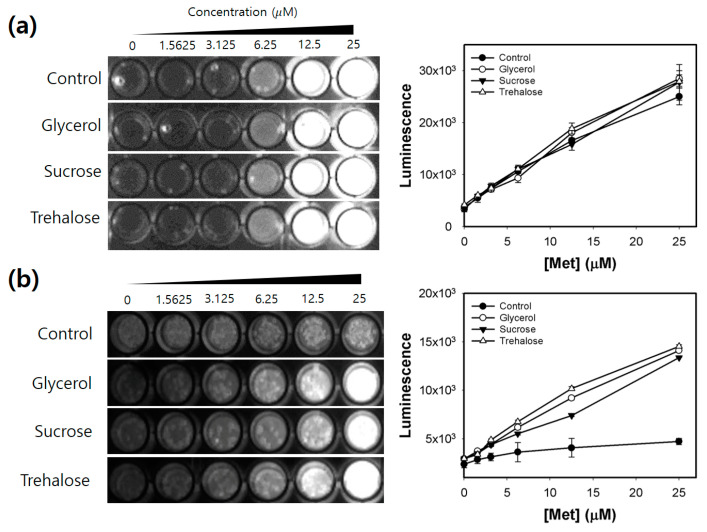
Luminescence images of the amino acid arrays containing glycerol, sucrose, and trehalose as a cryoprotectant and the calibration curves of methionine (Met) (plotted as concentration vs. luminescence intensity) generated using these arrays (**a**) before and (**b**) after storage for 3 months. Error bars represent standard errors derived from three independent measurements.

**Figure 6 biosensors-11-00252-f006:**
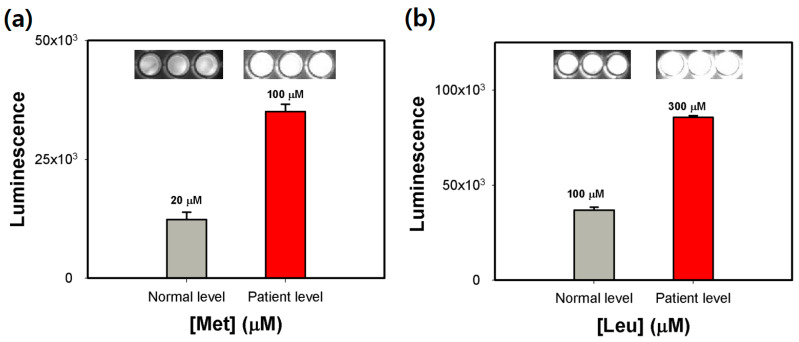
Diagnosis of (**a**) homocystinuria and (**b**) maple syrup urine disease using the ready-to-use amino acid array comprising Met and Leu auxotrophic *E. coli* strains with trehalose after storage for 3 months. Error bars represent standard errors derived from three independent measurements.

**Table 1 biosensors-11-00252-t001:** Performance of the ready-to-use Met, Ile, Lys, and Leu arrays to determine Met, Ile, Lys, and Leu concentrations, respectively, after storage for 3 months.

	Initial Concentration (μM)	Measured Concentration ^a^ (μM)	SD ^b^	CV ^c^ (%)	Recovery Rate ^d^ (%)
Met	10	10.4	0.35	3.47	100.83
Ile		10.7	0.28	2.71	103.37
Lys		9.4	0.47	4.79	97.63
Leu		10.7	0.55	5.41	100.93

^a^ Average value of five independent measurements. ^b^ Standard deviation (SD) of five independent measurements. ^c^ Coefficient of variation (CV) = (SD/mean) × 100. ^d^ Recovery rate = (Measured value/Expected value) × 100.

## Data Availability

Not applicable.

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
