# Peer review of "Effective Cryopreservation of a Bioluminescent Auxotrophic Escherichia coli-Based Amino Acid Array to Enable Long-Term Ready-to-Use Applications"

_biosensors, 2021, doi:10.3390/bios11080252_

Round 1

Reviewer 1 Report

This manuscript (biosensors-1304526) edited by Hee Tae Ahn et al. has determined effectiveness of several cryoprotectants in preservation ability of bioluminescent E. coli-based amino acid array to quantify amino acids in biological samples. The authors found that 5% trehalose is the most effective cryoprotectant under the storage condition (-20 ºC, 3 months) and this simple protocol showed that the auxotrophic E. coli-based amino acid array maintained linearity of target amino acid detection. Importantly, the cryo-array system has been shown to precisely detect clinically relevant levels of Met and Leu in patient level spiked blood serum samples.

The conclusions drawn from the results obtained are scientifically reasonable and clear, and the auxotrophic E. coli-based amino acid array system developed in this study can provide a platform for practical applications. However, this reviewer cannot evaluate whether the status of this new protocol is the final stage or developing stage, the final purpose of which is real applications of microbial cell-based biosensors. Are there any limitations of this stage of the system obtained in this study: e.g. three months storage is enough for commercial applications? If not, highlighting the limitations of the current study can provide readability and importance of this study.

Other specific comments are as follows:

P.3, L.110-114: Volume of cells and the seal materials have not been described. The information should be described.

P.6, L.205-206: The sentence describes about Figure 3(b). Thus, please add “Figure 3(b)” for readability.

P.7, Figure 4: Is the Y-axis “2e+4“ correct?

P.7, Figure 4 and 5: Plots in Fig. 5 at 25 µM in (a) and (b) are inconsistent with those of Fig. 4(a) at Day 0 and Day 90, respectively. How do the authors explain this discrepancy?

P.7, Figure 5 legend: Cryoprotectants, glycerol and sucrose, are missing.

Reviewer 2 Report

Suggestions to the Authors:

The authors presented a biosensor for analyzing amino acid in different fluids. The biosensor is based on an auxotrophic Escherichia coli strain and its demonstrated stability for storage over time. The work will significantly contribute to the field of biosensors and some revisions are required, mainly in terms of content clarification and paper's improvement.

1.- To evaluate the storage of this biosensor, the authors proposed 3 months of storage under determine conditions an analyzed the luminescence after this time. Were used any standard to compare the results ? For instance, other biosensors or sensor for this analysis. This point could be interesting for demonstrating the best viability of this sensor regarding the current technique.

2.- The authors did not indicate the standard deviation in figure 2 like they made in the rest of figures and they did not indicate how many replicas were carried out for obtaining these results. Please indicate this information y the adequate section.

3.- In section Results and lines from 145 to 149, the authors wrote “ In this study, an amino acid array comprising amino acid auxotrophic E. coli harboring a bioluminescence plasmid was constructed for rapid and convenient quantification of multiple amino acids…”. They must indicate the plasmid transform in E. coli strain for analysing the luminescence or the reference where this strain was constructed or if the strain used contain this plasmid. Moreover, the authors did not indicate the strain procedure (from some Type Culture Collection, for instance). Please, add the information request in the manuscript.

Reviewer 3 Report

This work demonstrated some cryoprotectants play a role in long-term preservation of the amino acid arrays that was based on bioluminescent whole cell biosensor and described a procedure of the ready-to-use amino acid array. This study is of significance for application of amino acid quantification in clinical diagnosis. However, this manuscript body should be revised and data in detail should be provided as supporting information.

1, The sentence “The clinical utilities of the Met and Leu array…….quantifying Met and Leu in blood serum samples for the diagnosis of corresponding metabolic diseases” in the abstract is not rigorous. Because authors applied the ready-to-use amino acid array with artifical blood serum samples, not the real blood samples according to the description in Page 8. Please revise the abstract.

2, Bioluminescence and its mechanism should be introduced in the “Introduction part” since this whole cell biosensor is based on bioluminescence.

3, Page 3, Line 98, I think “pTAC-luc” refers to the plasmid that encodes firefly luciferase according to the reference 15. My suggestion is that the cassette/gene about the bioluminescence function should be described clearly in methods.

4, Page 3, Part 2.3, firefly bioluminescence emits from luciferase-luciferin reaction. So substrate addition is needed in firefly bioluminescece assay. However, substrate and its concentration were not mentioned in the description of Part 2.3 or the other part of methods.

5, Page 4, Line 154, why choose these six cryoprotectants? Give an explanation in the manuscript.

6, Page 4, Line 179, why set 5% and 10% level of cryoprotectant? Did authors try other levels?

7, Page 5, Figure 2, more details about the treatment of data should be provided. My suggestion is that the original graph/table about bioluminescence and amino acid concentration should be illustrated as supporting information.

8, Page 6, Figure 3, the same suggestion as figure 2.

9, Page 9, Conclusions Part, discuss the reason why 5% trehalose is best for the preservation of biosenor array and possible mechanism.

10, Supplementary Materials Part, a procedure of ready-to-use amino acid array comprising E. coli auxotroph was depicted in supplementary Materials. However, the experimental procedure lacked the step of luciferin addition. The luciferin was not mentioned in main text or supplementary. Please check and revise.
